# A Protocol to Self-Familiarize Health Care Professionals with the Detection Limits of a Physical Activity Tracker for Low-Impact Steps in Patients Recovering from Knee Surgery—A Proposal and a First Evaluation

**DOI:** 10.3390/s25216666

**Published:** 2025-11-01

**Authors:** Werner Vach, Daniel Rybitschka, Scott Wearing, Andreas Gösele, Frances Weidermann, Marcel Jakob

**Affiliations:** 1Basel Academy for Quality and Research in Medicine, Steinenring 6, 4051 Basel, Switzerland; marcel.jakob@crossklinik.ch; 2Crossklinik AG, Bundesstr 1, 4054 Basel, Switzerland; 3TUM School of Medicine and Health, Technical University of Munich, Arcisstraße 21, 80333 Munich, Germany; s.wearing@tum.de; 4Department of Sport, Exercise and Health, Faculty of Medicine, University of Basel, Grosse Allee 6, 4052 Basel, Switzerland; 5Faculty of Medicine, University of Basel, Klingelbergstr. 61, 4056 Basel, Switzerland

**Keywords:** health care professionals, self-familiarization, knee surgery, physical activity tracker, post-surgical condition, step count

## Abstract

**Highlights:**

**What are the main findings?** 
A structured protocol can support health care professionals in mimicking the low-impact steps of patients recovering from knee surgery.Fourteen health care professionals could obtain some insights into the detection limits of four physical activity trackers using the protocol.
**What is the implication of the main finding?** 
Health care professionals may be able to familiarize themselves with the detection limits of a physical activity tracker using structured protocols.

**Abstract:**

Physical activity trackers are promising for monitoring physical activity in patients after surgery. However, the remobilization of patients following surgery is characterized by low-impact movements. It is often unclear to health care professionals whether a specific physical activity tracker is able to correctly detect steps in this patient population. A protocol is proposed, which allows health care professionals to familiarize themselves with the detection limits of a physical activity tracker. The professional should walk 20 steps under varying conditions mimicking the situation of patients after knee surgery. Conditions vary in step size, walking direction, use of walking aids, and footwear. The protocol was tested in a group of 14 health care professionals. Participants wore four trackers simultaneously, representing different modalities and different locations. For two trackers, the participants could experience a variation in the detection limits across the different conditions. On one hand, the within-participant reproducibility was substantial on average, though the between-participant reproducibility was only fair. On the other hand, experiencing incorrect step counts varied highly across and within participants. In conclusion, the self-familiarization of health care professionals with the detection limits of a physical activity tracker using specific protocols seems to be a feasible approach. Such protocols can provide valuable tools for facilitating the use of physical activity trackers in clinical applications. Additional research may allow for further refinement of the protocol to generate input that is more comparable across participants and closer to the gait of patients.

## 1. Introduction

After knee surgery, the treating clinician and other health care professionals (HCPs) are in charge of evaluating the individual course of a patient. Ensuring sufficient and adequate remobilization following surgery requires the clinician to accurately monitor the patient’s mobility and physical activity. However, it is difficult for HCPs to observe the mobility a patient has reached in daily life if they have to rely on patients’ self-reports. Such reports may be unreliable due to social desirability or cognitive impairments [1]. Physical activity trackers (PATs), based on accelerometers and other sensors, are promising to provide objective information concerning patient mobility [2], and validated user-friendly tools are commercially available [3,4]. Consequently, HCPs have, today, the opportunity to use PATs within their clinical routine, as well as for purposes other than monitoring. There already exist 39 systematic reviews on the intervention effects of PATs [5] and 15 studies on using commercially available PATs after total knee arthroplasty [6].

However, it is often unclear whether a specific PAT is indeed suitable for monitoring the changing level of mobility in a surgically treated patient. Early remobilization after surgery—in particular, after surgery of the lower extremities—may be characterized by slow, low-impact, and sporadic movements. In contrast, PATs have often been validated using healthy adults. Thus, they may simply fail to recognize low-level mobility. Indeed, a systematic review found that a specific tracker showed decreased validity when applied in unhealthy populations or at slow walking speeds [7]. There is an ongoing debate about the validity of PATs during early recovery [8,9,10,11,12,13,14]. Two studies have explicitly investigated the validity of different step counters at a walking speed of 0.4 m/s, and came to contradictory conclusions [15,16]. The use of walking aids seems to be an additional challenge for PATs [17,18,19]. Consequently, it is hard for HCPs to judge the information provided by a specific step count if there is some suspicion that not all steps are counted. Understanding the detection limits of a PAT helps to interpret step counts correctly. This also applies to information provided by a PAT beyond step counts, e.g., activity profiles based on a classification of physical activities, as most physical activities are achieved through short bouts of low-intensity activity.

A further issue arises from the continuous updating of a PAT’s algorithm by the manufacturer in order to improve the measurement process [20]. Hence, also for validated PATs, there is always some risk that their measurement properties have changed [21,22]—especially for unusual user groups, such as patients after surgery. Thus, a quick ad hoc validation is desirable.

It would be useful to have a simple protocol that can be used by HCPs to familiarize themselves with the detection limits for low-impact-steps of a specific PAT. Preferably, such a protocol should also take into account some additional conditions under which patients typically perform physical activities during the recovery phase. For example, many activities are performed indoor at home within a potentially cluttered environment or walking aids may be used. The protocol should be feasible for any HCP. In particular, it should be possible for HCPs to carry out the protocol without the need for additional equipment.

The aim of the current study is to propose and test such a protocol for the specific case of patients after knee surgery. The basic idea is to ask HCPs to wear the PAT, to perform steps similar to those they expect in patients during recovery after knee surgery, to vary the impact and other characteristics of the steps, and to always compare the number of steps they performed with the count shown by the PAT (cf. Figure 1). The basic approach to testing the protocol is to invite HCPs to carry out the protocol. This allows us to study its feasibility and whether HCPs can identify differences in detection limits across different conditions. In addition, the HCPs were asked to complete the protocol twice in order to study reproducibility.

## 2. Materials and Methods

### 2.1. The Protocol

The protocol consists of a series of conditions. Under each condition, the HCP performs twenty steps while wearing the PAT and aiming to mimic the gait of patients at varying stages after knee surgery. After each condition, the HCP inspects the number of steps counted by the PAT, which allows the HCP to make a comparison with the true number of steps.

The conditions are based on combinations of the following factors (Italic font indicates the abbreviations later used to refer to factor levels):
1Step size intended: 25%, 40%, 75%, 100% of normal step size.2Direction: Straight line, turn by 90° after 10 steps; zigzag line, 90°

shifts after 5, 10, and 15 steps; 360° circle clockwise.3Use of walking aids: Without walking aids; with walking aids4Footwear: Street shoes vs. home slippers.

The following 14 combinations are considered in specific order: S-100, S-75, S-40, S-25, T-75, Z-75, C-75, T-40, Z-40, C-40, W-75, W-40, L-75, and L-40.

In order to allow HCPs to get an idea of the intended changes in step size, a video illustrating the conditions S-100, S-75, S-40, and S-25 is provided (publicly available at https://osf.io/yrz7c (accessed on 29 October 2025)). It is recommended that HCPs watch this video and familiarize themselves to the conditions by performing some practice trials prior to executing the protocol. In addition, it is recommended that HCPs wear a bandage on one knee to remind them to perform the gait in a somewhat asymmetric manner.

### 2.2. Elaboration of the Protocol

Walking speed is well known to influence a number of gait-related variables, and its impact on a PAT’s ability to detect steps has been previously demonstrated [7]. Consequently, the first factor is aimed at controlling walking speed. As it is hard to directly standardize the speed of steps without external equipment, we decided to phrase the differences with respect to step size. Furthermore, low-impact steps typical for patients after knee surgery are not only characterized by reduced step sizes, but also by other aspects, such as reduced motor control, which manifest as clumsiness and heightened cautiousness. Consequently, we produced a video explaining the intended differences in speed and reminding users of the protocol to mimic the gait of patients after knee surgery.

In contrast to laboratory-based gait studies, in which steady walking on a straight line is usually examined, patients after knee surgery may perform the majority of their physical activities at home with obvious spatial restrictions. Hence, 20 steps taken in a straight line are unlikely to represent a typical behavior. Consequently, as the second factor, we consider a variety of deviations from a straight-line gait, which differ in the frequency and abruptness of the directional changes. These variants may have an additional impact on the detection properties of PATs to that of walking speed alone. Finally, two additional aspects are taken into account that are related to the recovery process—the use of walking aids—or to the home environment of the patients—the use of slippers. These constitute the third and fourth factors, respectively.

As a full factorial design covering all 64 possible combinations does not seem to be feasible, an incomplete factorial design is suggested. Regarding the combination of a straightforward direction at a step size of 75% or 40% with no walking aids and street shoes as the two core conditions, these two conditions are systematically varied with respect to one of the four factors, resulting in 14 conditions.

With respect to the sequence of the conditions, we suggest starting with S-100, reflecting “normal” gait in a patient, and to reduce the step size sequentially such that the HCP becomes more familiar with the step sizes also used later. We then focus on the three variants in walking direction at step size 75%, followed by the same three variants at 40% step size. We prefer to vary the direction in the first place, as we regard this as conceptually simpler than a change in step size. The placement of the four final conditions reflects the need to use specific equipment.

We do not specify the use of a specific walking aid. This way the HCP can choose the walking aid preferred in her or his patient population or which is at hand for the HCP. The four step sizes are not equidistantly chosen in order to have a sufficient difference in step size when considering only two step sizes while varying factors 2 to 4.

### 2.3. Study Population and Recruitment

Aiming at a protocol suitable for all HCPs in contact with patients after knee surgery, we included three different groups of HCPs, varying in the type of contact: orthopedic surgeons, physiotherapists, and sport scientists. All HCPs were required to have experience with managing patients after knee surgery.

Members of the project team—located at a multi-disciplinary clinic specialized in orthopedics and sport medicine—approached colleagues by email and invited them for participation. Interested HCPs were provided written study information and signed informed consents at the study visit. The study visits took place between 12 March and 26 June 2024.

### 2.4. PATs Used

HCPs were required to simultaneously wear four different PATs during the protocol. The four PATs selected are described in Appendix A, together with the criteria for their selection. They comprise a sensor integrated within a shoe *sole*, a sensor attached to the *knee*, a sensor which can be worn in the *trouser* pockets, and a sensor to be worn at the *wrist* (Appendix A). We refer to them in the following sections by their wearing location. They also involve different technologies to identify steps such as force-sensitive resistors, inertial measurement units, triaxial accelerometers, gyroscopes, magnetometers, temperature sensors, and barometers. Due to the differences in technology and wearing position, we expected differences in the ability to detect steps across the four PATs.

### 2.5. The Experimental Setup

Participants were invited to a single 90 min visit at the Biomechanics Lab at the crossklinik, Basel, Switzerland. Each participant was asked to wear their “usual” shoes and to bring a pair of slippers. The conduct of the study activities at the Biomechanical Lab was guided by varying instructors supported by varying assistants from the sport science group of the crossklinik. They had been partially involved in the development of the protocol.

The participants obtained detailed instructions and performed a few trial walks to become familiar with the tasks involved, i.e., using different step sizes, changing directions, performing a reading of the step counts from the different devices, using walking aids, etc. This included a 20-step walk in a straight line with a normal step size. From this walking distance, the assistant then calculated the corresponding distance for 25%, 40%, and 75% step size and marked them to support the later evaluation of whether the intended step size was reached. However, the marking was not communicated to the participant. Participants were asked to wear a bandage (Genumedi, E+Motion) on the right knee, while the knee sensor was placed on the left knee. The factor level “Turn” was implemented as a right turn, whereas the factor level “ZigZag” started with a left turn. Forearm crutches were used as a walking aid and the HCPs were asked to use a partial weight-bearing technique, which was also demonstrated by the instructor.

The trouser and the wrist sensors were set in a mode where steps were counted continuously. Before and after performing the steps under each condition, the participant read off the step count from these two sensors and communicated them to the assistant. The assistant computed the difference and communicated it back to the participant. The sole and knee sensors allowed the assistant to directly read off a step count from the accompanying app and to reset the app after each condition. The numbers were directly communicated to the participant. The two step count differences and the two step counts were recorded on a paper case report form (CRF) by the assistant.

The participants were asked to repeat the walk when the instructor identified that the performance of the task varied markedly from the experimental condition. The assistant counted, in any case, the number of steps performed by the participant and also recorded this on the paper CRF.

After performing the steps under all 14 conditions, the whole procedure was repeated in a second round. After the first round, the instructor asked the participants for the following information: age, gender, profession, and years of experience in working with patients after knee surgery. The information was recorded on the paper CRF. The participants were also asked to assess the value of the protocol and of the sensors by answering the following two questions separately for each PAT on a paper form:Do you now have an idea of the quality of the different sensors under the conditions used? (A precise idea/Some idea/A vague idea/No idea)Do you feel this PAT is suited to monitor the physical activity of patients after knee surgery? (This PAT is perfect/This PAT is well suited/This PAT is of limited value/This PAT is of no value)

### 2.6. Analytical Strategy

Although we aim at a fixed number of steps across all conditions, slight variations in the number of steps performed cannot be excluded. Hence most analyses will be based on the step count ratio, i.e., the ratio between the observed step count and true step count in a specific condition. A ratio close to 1 indicates that the PAT can detect a patient’s low-impact steps for this condition.

The inclusion of different conditions in the protocol aims at allowing the HCP to learn about the varying detection limits of the PAT. It is hence desirable to observe some systematic variation in the step count ratios across the conditions for a single PAT. Consequently, the first analytical step is to compare the distribution of the step count ratios—and, in particular, the mean values—across the different conditions. It is also of general interest to understand the influence of the different conditions on step counts, as this may reflect general limitations in assessing the steps of patients after knee surgery.

A good protocol should ensure that different HCPs come to a similar judgment about the same PAT. Hence, the HCPs should experience similar differences in step count ratios across the different conditions. Consequently, one outcome of interest is the inter-HCP reproducibility of step count ratios across the different conditions within the same PAT. As the sufficient intra-HCP reproducibility of step count ratios is a prerequisite for sufficient inter-HCP reproducibility, we first analyze the intra-HCP reproducibility before analyzing the inter-HCP reproducibility.

Potential participant effects on the step counts were assessed by visual inspection of the raw data. Finally, we report the distribution of the responses of the participants to the two questions about the value of the protocol and the value of the sensors.

### 2.7. Statistical Methods

*Raw data:* The raw data of the step count ratios were depicted by dot plots stratified by experimental condition and PAT, with the values of the two repetitions connected by a line. Participants were numbered in the order of their study visits.

*Effects of conditions:* The effect of conditions on the step counts was analyzed by a zero-inflated negative binomial regression model with the number of true steps as exposure and the experimental conditions as the only covariate. This model takes into account that conditions may influence the detection limit—implying, potentially, that no steps can be detected—and the ability to count steps above the limit correctly to a different degree and that participant effects introduce heterogeneity in counts. A potential dependence between different step count ratios within one HCP was taken into account by basing statistical inference on the Huber–White sandwich estimator [23,24]. Selected contrasts for marginal means (based on setting the true step count to 20) were considered to assess the effect of the step size within the categories S, T, Z, C, W, and L and the difference between category S and categories T, Z, C, W, and L, respectively. In addition, the *p*-value of testing the null hypothesis of no difference between all conditions was reported. A 5% statistical significance level was used.

*Reproducibility of step counts—general considerations:* Regarding step count ratios as quantitative measures, the reproducibility can be described by the intra-class correlation coefficient (ICC). However, in this study, step count ratios aimed at distinguishing three possible outcome scenarios for a specific experimental condition and a specific PAT:The PAT is able to identify steps under this condition. In this case, we expect a ratio close to 1.0.The PAT is unable to identify steps under this condition. In this case, we expect a ratio close to 0.0.The experimental condition is close to the limit at which the PAT can detect step counts. In this case, we have to expect the step count ratio to be rather unstable and may cover a rather wide range of values.

Consequently, ICCs are limited with respect to catching the reproducibility of interest, as they expect reproducibility over the whole scale from 0.0 to 1.0. To address this issue, we consider also weighted agreement rates. In determining the degree of agreement between two single step count ratios, the following weights were used:0.0—If one step count ratio is above 0.8 and the other is below 0.2;0.5—If one step count ratio is between 0.5 and 0.8 and the other is below 0.2, or if one step count ratio is between 0.2 and 0.5 and the other is above 0.8;1.0 If both step count ratios are between 0.2 and 0.8, or if one step count ratio is above 0.8 and the other above 0.5, or if one step count is below 0.2 and the other is below 0.5.

The choice of the thresholds 0.2 and 0.8 reflects that even true ratios of 0 or 1 might be affected by some noise. The agreement rate of a set of a pair of step count ratios was then defined as the average weight over all pairs. As pointed out in Appendix A, such agreement rates can also be transformed into κ values following the principle introduced by Cohen [25]. In verbalizing the magnitude of these κ values, we applied the classification into poor, slightly, fair, moderate, substantial, and almost perfect using the cut-off values 0.2, 0.4, 0.6, and 0.8 suggested by Landis and Koch [26].

*Reproducibility of step counts:* For each participant and PAT, the intra-HCP reproducibility was assessed by a kappa value and an ICC. These were based on all pairs of step count ratios observed for the same condition across the first and second round in one HCP (Figure 2). For each PAT, the inter-HCP reproducibility was assessed by a kappa value and an ICC. These were determined by computing these values for each pair of HCPs and each combination of rounds using all pairs of step count ratios observed for the same condition and averaging over all HCP pairs and over all combinations of rounds (Figure 2).

Computation of the ICC was based on a random effect model with the conditions as random effects using the restricted maximum likelihood (REML) technique. The reporting was omitted if the standard deviation of the random effect was less than 0.1. Reporting of kappa values was omitted if all step count ratios were above 0.5.

*Statistical software:* All computations were performed with Stata 17.1. The code for the main analyses is documented in Appendix A.

*Sample size:* In the study protocol, two different scenarios varying in the expected agreement rate between two HCPs were simulated and the precision of the estimates of the agreement rate were compared. It was concluded that 16 HCPs would be sufficient to distinguish the two scenarios.

## 3. Results

### 3.1. Study Population

Fourteen HCPs participated in the study. Two additional HCPs who originally consented to participate were unable to attend the planned study visits due to time constraints. Characteristics of the 14 participants are shown in Table 1. Five male and one female surgeon, all above 40 years of age with at least 15 years of experience, were included. Similarly, two male and two female physiotherapists, ranging in age from 24 to 42 years with 2 to 19 years of clinical experience were included. In addition, three female and one male sport scientist, all below the age of 26 years with up to 4 years of experience, completed the study protocol.

### 3.2. Raw Data

Appendix A depict the observed values of the step count ratios. Application of the knee sensor failed in 9 out of the 14 participants due to technical problems with the accompanying app, such that no data was produced. In the first participant, the target step sizes could not be achieved in the first round and the instructor decided not to record the data. In addition, the instructor requested 19 out of 350 trials to be repeated. Only four participants brought slippers to the visits, and one used them only in the first round. The other participants reported that they did not routinely wear specific slippers at home or simply do not wear any shoes at home. Due to the limited amount of data for the slipper condition, this condition was omitted from all analyses.

### 3.3. Distribution of Step Count Ratios and Impact of Conditions

Figure 3 depicts the distribution of the step count ratio values for each sensor under each condition. For the sole sensor, all distributions are restricted to a small range of values around 1, and this holds also nearly for the knee sensor for the five patients providing data. For the trouser and the wrist sensors, a wide range of ratio values, ranging from 0 to more than 1.5, could be observed.

The trouser sensor showed distinct differences between the conditions (*p* < 0.001). An analysis of the contrasts of interest (Appendix A) revealed that there was a statistically significant decrease with decreasing step size for all categories and the categories Z and W differed significantly from S with lower mean values. The wrist sensor showed much less pronounced differences between conditions than the trouser sensor, but they were still statistically significant (*p* < 0.001). The association between step size and step count ratio measured by the wrist sensor was significant for S, C, and W. However, none of the categories C, T, Z, and W differed significantly from S.

The very limited variation in step count ratio values for the sole and the knee sensor imply that these sensors cannot contribute to a quantitative assessment of reproducibility in a meaningful manner. Consequently, analyses of reproducibility are restricted to the knee and the wrist sensor in the following section.

### 3.4. Intra- and Inter-Rater Reproducibility

The intra-rater reproducibility of each participant across the 14 conditions for the trouser and the wrist sensors is depicted in Figure 4. For the trouser sensor, the average kappa value was 0.62, indicating substantial agreement. Only 3 out of 13 participants failed to reach a kappa of 0.4, i.e., at least a moderate agreement. The average ICC was 0.55 and four participants reached an ICC of above 0.7. There was substantial variation across participants, which coincides with the visual impression in Figure 1: some HCPs could fairly reproduce ratio values (e.g., 4, 5, 13, and 14), whereas others failed to do so (e.g., 2 and 10).

For the wrist sensor, the variation was too limited to allow for the computation of kappa values in seven participants. In the remaining six participants, the average kappa value was 0.44, and three participants reached a kappa value of above 0.4, i.e., indicating moderate agreement. ICC values were above 0.7 in two out of eight participants, with an average of 0.52.

The inter-rater reproducibility is depicted in Table 2. For the trouser sensor, a kappa value of 0.38 was observed, i.e., only a fair degree of agreement. The ICC was estimated as 0.45. For the wrist sensor, the results were even worse.

### 3.5. Participant Effects

For the wrist sensor, the visual inspection of the raw data (Appendix A) clearly indicates participant effects on the step counts. For some participants, the step count ratios were rather close to 1.0 for all conditions and over both rounds. For other participants, there was a substantial variation in the ratios over the conditions with a pattern varying from participant to participant, which could or could not be reproduced by the participant.

For the trouser sensor, only one participant was able to generate ratios close to 1 for nearly all conditions over both rounds.

### 3.6. Value Assessment by the Participants

With respect to the value of the protocol experienced by the participants, all of them agreed to have a precise idea about the sole sensor (Table 3). This also holds for the knee sensor among the five participants experiencing this sensor. With respect to the trouser sensor, the opinions were widely varying, and with respect to the wrist sensor, the majority felt to have had some idea.

With respect to the value of the sensors themselves, all participants expressed a favorable opinion with respect to the sole sensor. This also holds for the knee sensor among the five participants experiencing this sensor. All participants expressed an unfavorable opinion with respect to the trouser sensor. With respect to the wrist sensor, the opinions were more mixed.

None of the participants reported any complaints or difficulties with the protocol and there were no adverse events.

## 4. Discussion

### 4.1. Summary of Main Results

The protocol turned out to be feasible for all participating HCPs. For two PATs, the protocol allowed the HCPs to experience differences in the step count ratio across different conditions.

The sole and the knee sensors performed nearly error-free over all conditions, and this result was shared among HCPs. This also resulted in a rather uniform, favorable judgment about the value of these two sensors. Due to technical problems, however, the knee sensor could only be experienced by 5 of the 14 HCPs.

The trouser sensor was the only device that showed clear differences in the average step count ratios across the conditions. Hence, only for this sensor did we reach a favorable precondition to assess the value of the protocol by considering reproducibility for all participants. We observed, on average, a substantial intra-rater reproducibility, suggesting that essential differences between step count ratios can often be reproduced when repeating the protocol. The inter-rater reproducibility was only fair, indicating some limitations of the protocol with respect to ensuring comparable results across HCPs. However, it is notable that all HCPs were in agreement in providing an unfavorable judgment about the value of the sensor.

With respect to the wrist sensor, the moderate degree of systematic differences between the conditions limited the possibility of investigating the reproducibility in a systematic manner. However, a simple visual inspection of the raw data indicated clear participant effects on the step counts: some participants experienced step count ratios close to 1.0 over all conditions in both rounds, whereas other experienced a substantial variation across conditions and rounds. This variation in experiencing variation across the conditions may also explain the mixed opinions about the value of this sensor.

### 4.2. Implications with Respect to the Value of the Protocol

In general, our study confirmed that the protocol has the potential to inform HCPs about the utility of the PATs in detecting specific types of low-impact steps that can be expected in patients after knee surgery: it allows HCPs to observe corresponding differences in step counts for a PAT across the conditions considered—if they exist.

It was our aim to develop a protocol with a sufficient standardization of the input presented to the PATs, such that HCPs can come to similar conclusions about the detection limits of a PAT and its general value. This was the case for the sole and the knee sensors—which is, however, a highly trivial result as the sensor could detect nearly all steps. We were also, to some degree, successful with respect to the trouser sensor with a substantial intra-observer reproducibility on average and nearly moderate inter-observer variability. However, we failed with respect to the wrist sensor with clear differences between HCPs.

When trying to explain the difference between the trouser and the wrist sensors, the paired design of the study is a key element: the steps produced by the HCPs—and actually any movement of the body—were exactly the same. Hence, from this perspective, the “input” to both PATs was the same. Consequently, the wrist sensor must have been sensitive to some information in the input representing noise (relative to the information on the steps), e.g., to other types of body movements we did not control by our instructions. This would indicate an insufficient step counting quality of the sensor in this specific clinical context and not a failure of our protocol. This is in line with our observation that participants could experience substantial variation in step count ratios although the systematic differences across the conditions were small.

It can also be argued that, from the perspective of a single HCP, the main purpose of varying conditions is to generate some meaningful variation in the HCPs gait and to allow the HCP to experience how his or her choice of gait influences the ability of the PAT to detect steps. This aim is still reached even if the interpretation of the instructions and hence the input to the PATs differs from HCP to HCP. On the other hand, it is additionally desirable that HCPs experience at least roughly the same condition in the same manner. If HCPs differ in their impressions about which type of steps a PAT can detect, they will also differ in the interpretation of an activity profile generated by the PAT, and hence potentially in the consequences in managing a patient.

### 4.3. Potential Improvement of the Protocol

The limited inter-observer variability calls for some measures to further standardize the input to the PATs generated by the HCPs. The use of auditory or visual cuing to control step size and step frequency may be an option. Such approaches may, however, influence other gait parameters and may distract the HCPs from the task of mimicking the gait pattern of patients. Virtual or augmented reality [27,28] or biofeedback [29] may overcome this drawback. In addition, it might be possible to increase the quality of the input with respect to mimicking patients after knee surgery. Instead of a simple bandage, more specific means may be used, e.g., braces or orthotics, or patients could be used in generating the instructional video.

There still remains a need to identify further conditions which may reflect challenges for a PAT to identify steps correctly and which are likely to occur during the recovery period of patients after knee surgery. This can be, for example, other types of walking aids such as walking frames or supporting persons, performing half steps instead of full steps, walking upstairs or downstairs, or to differentiate between open-heeled footwear and closed footwear. Simulating a complete home environment to study the measuring properties of PATs has been recently suggested [14].

It might also be of interest to extend the protocol by allowing or forcing the HCPs to repeat conditions. This way, we can diminish the risk of false conclusions due to some unintended random variation in the input. In addition, this way, HCPs can take the condition-specific reproducibility of step counts into account in generating an opinion about the reliability of a PAT.

### 4.4. General Aspects of the Value of the Protocol

It constitutes a basic limitation of the protocol that it cannot be applied to all conceivable PATs. HCPs are typically interested in monitoring patients over time periods of several days and weeks. This may require the pre-processing of measured data and storing relevant information with low sampling frequency. It might be impossible to read off step counts after 20 steps under such conditions. This is the case, for example, with the Active Insights Band (https://activinsights.com/technology/activinsights-band (accessed on 29 October 2025)), which was especially developed for HCPs.

An advantage of the protocol is that it can be easily adapted by the HCP to meet specific needs or interests. If an HCP has concerns regarding specific gait patterns, corresponding conditions can be simply added to the protocol. The protocol can also be easily adapted to other patient groups, e.g., patients after hip surgery, abdominal surgery, or poly traumata, by generating corresponding videos and adjustments of conditions.

As with any experimental protocol applied to humans, potential risks have to be balanced against the gain in information. Rapid turns, zigzags, and the use of crutches is not without risk. However, as HCPs constitute the intended user group, it seems fair to expect that users can balance risk vs. information gain individually.

### 4.5. Further Aspects

Our study confirms that the impact energy of steps can have a substantial influence on the detection of step counts by a PAT. In addition, deviations from moving straight forward, which commonly occur in non-laboratory environments, can influence the performance of a PAT to detect steps. The use of crutches resulted in an average decrease of eight steps counted by the trouser sensor, but no such effects could be observed for the other PATs. This underlines that the effect of walking aids on the detection of steps is device-specific, and the use of crutches may not necessarily constitute a problem.

A nearly optimal overall accuracy indicates that the sole sensor would seem an expedient candidate for further investigation of its usefulness in monitoring the physical activity level of patients. However, the sensor is predicated on the concomitant use of footwear, which may represent a practical limitation to its application in the home environment, where footwear may not be routinely used. A high overall accuracy could also be observed for the knee sensor. Technical problems with the accompanying app do not necessarily reflect an obstacle for its use, as the sensor offers two different modes for short-term and long-term observations.

The inclusion of a popular fitness tracker (the wrist sensor) allowed us to address the potential wish of patients to use their own tracker to monitor post-surgical rehabilitation. For some HCPs, the sensor failed to detect steps under some conditions. This suggests some risk associated with the use of such a tracker, as it may not work well for some patients. This is in line with a reported average relative error rate of more than 40% for step counts from a 100 ft walk in patients after knee surgery when using two popular fitness trackers [9].

With a few exceptions, the sole sensor consistently underestimated the true number of steps by one step, independent of whether the participant succeeded in 20 steps or not. This probably reflects that there was always a half step at the start and at the end of the 20 steps.

### 4.6. Limitations of the Study

It was a distinct limitation of this study that only one of the four PATs tested showed clear differences between the different experimental conditions and that two of the sensors worked perfectly for all conditions. This prohibited further analyses planned in the study protocol, in particular, the usefulness of conditions to distinguish between PATs and the similarity of conditions in this respect. This would have allowed us to develop suggestions for removing conditions providing similar information.

The failure to include the slipper condition implies a risk of overlooking a potential challenge to the PATs and the limited amount of data for the knee sensor may have prevented further insights from a high-quality, but still imperfect, sensor.

The limited number of participating HCPs and the distinct association between the four HCP characteristics assessed prohibited an investigation of the latter with respect to a potential influence on step count ratios and inter-rater reproducibility.

It cannot be excluded that the variation in instructors across the participants has contributed to the limited inter-HCP reproducibility. Similarly, the assessment of the ground truth by the assistant need not be error-free, contributing to the underestimation of reproducibility.

The intended sample size of 16 HCPs was not reached. However, the sample size consideration aimed at ensuring that moderate differences between PATs with respect to detection limits could be demonstrated. Fortunately, the differences were rather distinct, and hence 14 HCPs were sufficient to obtain insights.

It can also be seen as a limitation that the participants were asked about their opinion on the value of the protocol and of the sensor only once after the completion of the first round. This choice reflected the wish to catch the intended use in practice, i.e., a single application of the protocol. An additional interrogation prior to executing the protocol would have allowed the assessment of a change in opinion, while an additional interrogation after the second round would have allowed the assessment of the effect of experiencing a repetition.

### 4.7. Outlook

Our investigation illustrates the feasibility of self-familiarizing health care professionals with the detection limits of a physical activity tracker using specific protocols. At the moment, the proposed protocol has to be used with some care, as it cannot be ensured that all HCPs will come to the same conclusion for all PATs. We recommend trying to reproduce findings when using the protocol in order to judge their reliability. In addition, further research is necessary regarding how individual variations in gait can influence the detection of steps by PATs. This may help us to understand their general usefulness in this patient population and to improve the protocol. Corresponding studies should implement a stricter control of conditions and additional data collection (e.g., video recordings) than the present study, which focused on a proof of principle.

The idea of self-familiarization of HCPs with PATs before offering them to their patients is a rather general one. The use of an experimental protocol to generate the input to PATs is only one possibility. Alternatively, real-world input can be used. For example, to investigate the validity of daily or weekly activity profiles, HCPs may wear the sensors themselves for some days and compare the recorded profile with the activities they actually did.

The lack of concrete protocols to facilitate the clinical application of PATs has been identified as a major barrier to their use to improve patient care [30]. The development of protocols for self-familiarization can make a contribution to enhance the clinical use of PATs and supplement existing guidelines for the implementation of PATs in clinical practice [31].

## 5. Conclusions

The type of protocol presented has the potential to inform HCPs about the ability of a PAT to detect low-impact steps that are characteristic of patients following knee surgery. The concept of self-familiarization may allow for the development of valuable tools to facilitate the clinical application of PATs. Further research should focus on optimization of the protocol with respect to generating input more comparable across HCPs and closer to the gait of patients.

## Figures and Tables

**Figure 1 sensors-25-06666-f001:**
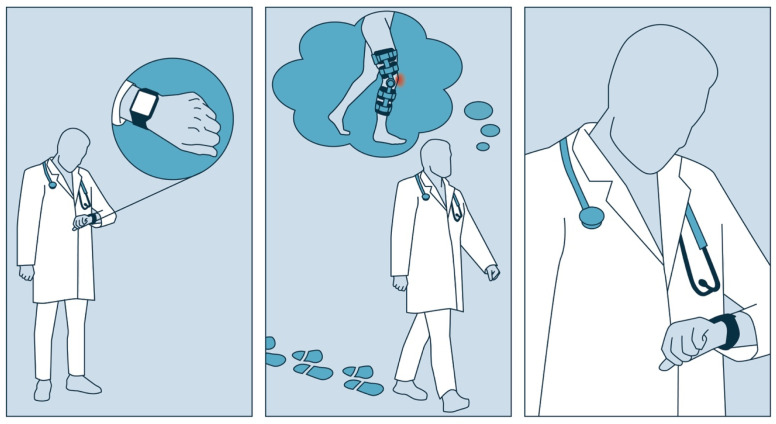
A schematic visualization of the basic approach of the protocol.

**Figure 2 sensors-25-06666-f002:**
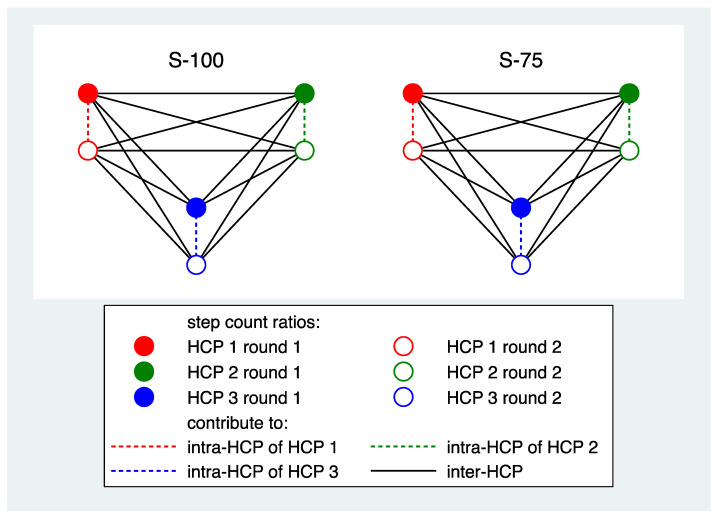
A schematic presentation of the contribution of the data of three HCPs and two conditions to determine intra- and inter-HCP reproducibility for one PAT.

**Figure 3 sensors-25-06666-f003:**
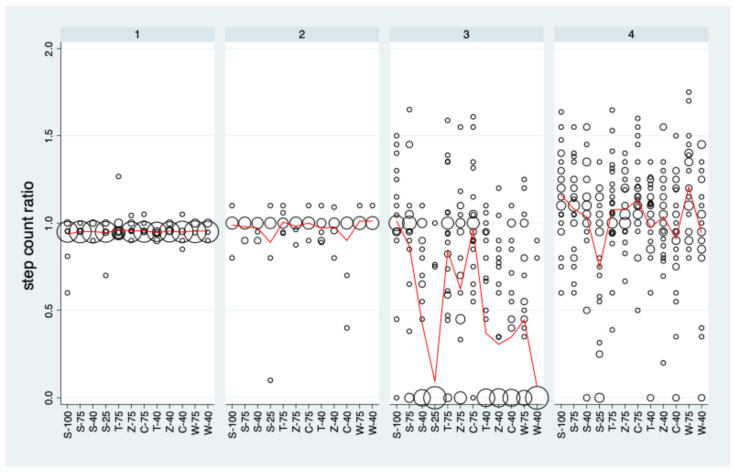
The distribution of the step count ratios stratified by condition and PAT. Data are pooled over rounds 1 and 2. Bubble sizes (areas) are proportional to the number of observations with the specific step count ratio. The red curves represent the mean values.

**Figure 4 sensors-25-06666-f004:**
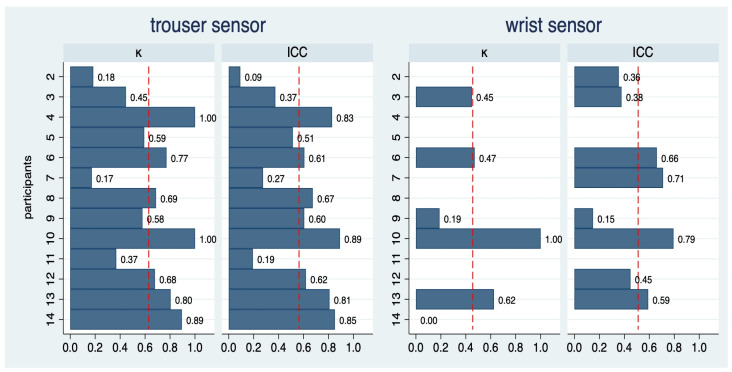
Intra-rater reproducibility of step count ratios for 13 of the 14 participants for two sensors. As data from the first round of participant 1 was not usable, this participant was not included in this analysis. The dashed red vertical lines indicate the average value across all participants. For the wrist sensor, the computation of kappa values was restricted to 7 participants and the computation of ICC values to 8 patients.

**Table 1 sensors-25-06666-t001:** Characteristics of the study participants. Experience is expressed in years.

	Health Care Profession	Age	Sex	Experience
1	Surgeon	42	f	15
2	Surgeon	45	m	15
3	Surgeon	80	m	40
5	Physiotherapist	31	f	3
6	Physiotherapist	26	m	8
7	Sport scientist	24	f	1
8	Surgeon	59	m	30
4	Physiotherapist	42	m	19
9	Surgeon	60	m	20
11	Surgeon	51	m	15
10	Sport scientist	25	f	1.5
12	Physiotherapist	24	f	2
13	Sport scientist	26	m	4
14	Sport scientist	26	f	3

**Table 2 sensors-25-06666-t002:** Inter-rater reproducibility of step count ratios for two sensors.

	κ	ICC
trouser sensor	0.38	0.45
wrist sensor	0.09	0.34

**Table 3 sensors-25-06666-t003:** Distribution of the responses of the patients to two questions. Absolute frequencies are given. The knee sensor could be evaluated only by 5 participants.

Do you now have an idea of the quality of the different sensors under the conditions used?				
	sole	knee	trouser	wrist
A precise idea	14	5	3	0
Some idea	0	0	3	10
A vague idea	0	0	5	3
No idea	0	0	3	1
Do you feel this PAT is suited to monitor the physical activity of patients after knee surgery?				
	sole	knee	trouser	wrist
This PAT is perfect	13	3	0	0
This PAT is well suited	1	2	0	5
This PAT is of limited value	0	0	5	8
This PAT is of no value	0	0	9	1

## Data Availability

The experimental data generated in this study is included in the Appendix A. Further inquiries can be directed to the corresponding author.

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
