# Peer review of "A Protocol to Self-Familiarize Health Care Professionals with the Detection Limits of a Physical Activity Tracker for Low-Impact Steps in Patients Recovering from Knee Surgery—A Proposal and a First Evaluation"

_sensors, 2025, doi:10.3390/s25216666_

Round 1

Reviewer 1 Report

Comments and Suggestions for Authors

This study proposes and initially validates an experimental protocol for health care professionals (HCPs) to self-familiarize themselves with the detection capability of physical activity trackers (PATs) during low-impact gait, specifically designed for the rehabilitation phase following knee surgery. The reviewer has the following issues for the authors to consider.

  1. The study reports that the knee sensor failed to generate data in 9 out of 14 participants due to app malfunctions. Additionally, only 4 participants took part in the slipper condition, which was ultimately excluded from analyses. While these data missing issues are mentioned in the Results section, the discussion on their implications remains superficial. The authors are advised to further elaborate on the impact of this drastic reduction in sample size on statistical power and the robustness of the conclusions.
  2. Results indicate low consistency among participants under the same gait conditions. Although the authors mention the potential use of cuing methods for improvement in the Discussion section, they do not explicitly analyze whether such inconsistencies affected the sensor results. Pleaseaddress this point and give more explanations.
  3. The authors repeatedly emphasize that the protocol is ‘ready for use’ in the Abstract, Discussion, and Conclusion sections. This expression can easily cause confusion, please consider modifying it.
  4. The authors attribute the poor performance of the wrist sensor to noise interference from arm movements but do not explore other potential factors. Pleaseprovide a more comprehensive analysis of the wrist sensor results.
  5. Figures 3 and 4 illustrate the step count ratio distribution and consistency of different sensors, yet the text provides only a brief explanation of the trends and differences in these figures. The authors should further analyze the data characteristics therein to help readers better understand the phenomena reflected by the figures.

Author Response

Comment 1: The study reports that the knee sensor failed to generate data in 9 out of 14 participants due to app malfunctions. Additionally, only 4 participants took part in the slipper condition, which was ultimately excluded from analyses. While these data missing issues are mentioned in the Results section, the discussion on their implications remains superficial. The authors are advised to further elaborate on the impact of this drastic reduction in sample size on statistical power and the robustness of the conclusions.

Response 1: The section about limitations was extended and is now additionally covering the following points:

“The failure to include the slipper condition implies a risk of overlooking a potential challenge to the PATs and the limited amount of data for the knee sensor may prevented further insights from a high quality, but still imperfect, sensor.”

“The intended sample size of 16 HCPs was not reached. However, the sample size consideration aimed at ensuring that moderate differences between PATs with respect to detection limits could be demonstrated. Fortunately, the differences were rather distinct, and hence 14 HCPs were sufficient to obtain insights.”

Comment 2: Results indicate low consistency among participants under the same gait conditions. Although the authors mention the potential use of cuing methods for improvement in the Discussion section, they do not explicitly analyze whether such inconsistencies affected the sensor results. Please address this point and give more explanations.

Response 2: As all participants were exposed to the same type and amount of cuing, it is unfortunately not possible to analyse the effect of variation in cuing.

Comment 3: The authors repeatedly emphasize that the protocol is ‘ready for use’ in the Abstract, Discussion, and Conclusion sections. This expression can easily cause confusion, please consider modifying it.

Response 3: We followed the advice of the reviewer. We avoid this term and focus instead on having demonstrated the general feasibility of the idea of self-familiarization.

The corresponding sections are now:

Abstract, Conclusion:

“In conclusion, self-familiarization of health care professionals with detection limits of a physical activity tracker using specific protocols seems to be a feasible approach. Such protocols can provide valuable tools to facilitate the use of physical activity trackers in clinical applications. Additional research may allow further refinement of the protocol to generate input more comparable across participants and closer to the gait of patients.”

Outlook, first sentence:

“Our investigation illustrates the feasibility to self-familiarize health care professionals with detection limits of a physical activity tracker using specific protocols.”

Conclusion at the end of the paper:

“The type of protocol presented has the potential to inform HCPs about the ability of a PAT to detect low-impact steps that are characteristic of patients following knee surgery. The concept of self-familiarization may allow to develop valuable tools to facilitate the clinical application of PATs. Further research should focus on optimization of the protocol with respect to generate input more comparable across HCPs and closer to the gait of patients.”

Comment 4: The authors attribute the poor performance of the wrist sensor to noise interference from arm movements but do not explore other potential factors. Please provide a more comprehensive analysis of the wrist sensor results.

Response 4: With respect to the poor performance of the wrist sensor, we feel that we have the strong point of paired observations. “Execution” was identical for the wrist and the trouser sensor, as they were worn together. There may be some aspects specific for the wrist sensor, for example arm movements. However, we do already use the more general term “body movements” to indicate that it can go beyond arm movements.

It was not the aim of the study to investigate the behaviour of the sensors in detail. For this reason, no additional data (e.g. based on video recordings) was collected. Hence, we cannot provide a more detailed analysis of the wrist sensor results looking for causal explanations.

However, as pointed out in the Discussion, our results are in line with those of other investigations.

Comment 5: Figures 3 and 4 illustrate the step count ratio distribution and consistency of different sensors, yet the text provides only a brief explanation of the trends and differences in these figures. The authors should further analyze the data characteristics therein to help readers better understand the phenomena reflected by the figures.

Response 5: It is the main purpose of Figure 3 to inform the reader about the existence of some systematic differences between the conditions – as this is a prerequisite for the following analyses of inter-rater reproducibility. Without differences between the conditions, there is nothing to reproduce.

We tried to clarify this point by moving the first paragraph of Section 3.4 up to the end of Section 3.3, such that the reader can understand the reason for the lack of a more detailed analysis. (A study focusing on the differences between the conditions would also require a more stringent design.)

Figure 4 represents simple the raw data for the inter-observer variation analysis, i.e. the values behind the mean values finally reported in the Results section. To clarify this, we added the mean values in Figure 4 and report also the mean value of the ICC values when discussing the results for the trouser sensor.

Reviewer 2 Report

Comments and Suggestions for Authors

 Reviewer's comments

Comment 1 (p. 3). Let the figures carry more weight. Add confidence intervals or standard errors and consider a glimpse of subject-level traces so readers can see spread and outliers. A quick sensitivity check that drops mis-executed trials would also build trust.

Comment 2 (p. 3; also p. 9, p. 15). “Ready for use” reads a bit ahead of the data. You relied on clinicians imitating patients, and several planned conditions/sensors didn’t survive. Those two limitations should temper how broadly you pitch the conclusions.

Comment 3 (p. 4). Step size isn’t the same as “low-impact.” Impact is about cadence, velocity, and forces, not just geometry. Without direct cadence/speed or force thresholds, the label feels assumed rather than measured.

Comment 4 (pp. 4–5). Running everyone through a fixed sequence invites learning, fatigue, and expectation effects. Randomization or counterbalancing would help separate true condition effects from order artifacts.

Comment 5 (pp. 4–5). The turning and zig-zag tasks are underspecified. Path width, turning radius, and allowable angular speed matter; without them, between-person variability can swamp the contrasts you want to study.

Comment 6 (p. 4). The paper drifts from step counting into “activity profiles,” but the protocol never measures posture or non-ambulatory behavior. Tightening the scope to what you actually recorded would make the claims cleaner.

Comment 7 (pp. 4–6). Safety and ethics deserve a clearer line of sight. Rapid turns, zig-zags, and crutch use in a corridor have non-zero risk; a brief note on monitoring, near-misses, or adverse events would reassure readers.

Comment 8 (p. 5). Multiple instructors and assistants add procedural noise you can’t model away. Either standardize more tightly or acknowledge and analyze rater effects; otherwise device differences may reflect who ran the trial.

Comment 9 (p. 5). The crutch condition is handled qualitatively—“partial weight bearing”—with no control of load, cadence, or placement. That makes the “walking-aids” factor hard to interpret: differences could be execution, not device.

Comment 10 (p. 6). Ground truth rests on one assistant’s manual count with no independent check. Even a light reliability pass (second rater or video spot-checks) would raise confidence in the reference.

Comment 11 (p. 6). Reading totals before/after and differencing is fragile. Transcription slips, display lag, or reset issues happen; describe checks to catch them or move to an automated log to harden the chain.

Comment 12 (pp. 6–7). The analysis hops between counts and ratios without a unifying plan. If ratios are the target, use models built for bounded outcomes (e.g., beta or hurdle) instead of forcing everything through a count model.

Comment 13 (p. 6). Protocol fidelity isn’t quantified. How often were targets missed, by how much, and how were those trials handled? Without that, it’s tough to separate condition effects from plain execution error.

Comment 14 (p. 6). Condition effects (e.g., crutches reducing trouser-sensor counts) don’t account for subject-level variability. A simple mixed model would help distinguish device behavior from individual strategy.

Comment 15 (p. 6). The protocol assumes devices show immediate readable counts, which many trackers don’t. That practical constraint narrows applicability more than the text lets on—worth stating plainly.

Comment 16 (p. 7). The choice of a zero-inflated negative binomial for ~20-step trials is under-argued, and diagnostics are missing. Show why ZINB is preferred and that it fits, not just that it runs.

Comment 17 (p. 7). With many condition/category contrasts, some multiplicity control is expected. Otherwise a few “significant” hits may just be noise; even a transparent trade-off discussion would help.

Comment 18 (p. 7). Agreement categories are borrowed and widely debated, and the metric lives mostly in the supplement. Define it clearly in the main text and justify thresholds with task-relevant reasoning.

Comment 19 (p. 7). Data sharing is good, but there’s no pre-specified analysis plan or code. Given custom metrics and nonstandard modeling, a small, documented workflow would make replication real.

Comment 20 (p. 9; also pp. 14–15). The conclusion leans toward endorsing a single sensor despite dropped conditions and failures elsewhere. If you keep that message, frame it as provisional and bound by the observed gaps.

Comment 21 (p. 9). Execution problems—knee-sensor failures, the missing slippers condition—aren’t minor. They change design balance and interpretability, and the discussion should grapple with that directly.

Comment 22 (p. 10). Reproducibility concerns may be execution, not device. Without cadence or step-length checks, you can’t localize the instability, so claims about device inconsistency should be softened.

Comment 23 (pp. 10, 14). Clinical readiness is over-promised. Agreement is modest and inputs vary across people and conditions; “promising method development” is a better fit than “ready for use.”

Comment 24 (pp. 10, 15). The sample-size story is thin: the target wasn’t met, precision goals aren’t stated, and heterogeneity is high. Emphasize interval estimates and acknowledge limited power.

Comment 25 (p. 14). Key device details are missing—firmware, sampling rates, filters, algorithm modes. Those parameters move step counts in practice; without them, comparisons and replications are guesswork.

Author Response

We are very grateful to the reviewer for the detailed and constructive comments.

Comment 1 (p. 3). Let the figures carry more weight. Add confidence intervals or standard errors and consider a glimpse of subject-level traces so readers can see spread and outliers. A quick sensitivity check that drops mis-executed trials would also build trust.

Response 1: We completely agree with the reviewer that confidence intervals can add valuable information. However, they make mainly sense if they refer to single numbers with a high impact on the interpretation of the results. From this perspective, it would be desirable to have some confidence intervals in Section 3.4, presenting the main quantitative results. However, it is unclear how to compute reliable confidence intervals in such a small sample setting with highly non-normal data for non-standard quantities such as average kappa or average ICC values. Hence, we decided to make no attempt in this direction.

We cannot see a good reason to add confidence intervals in Figure 3 and Figure 4. For both figures it holds that it is the overall pattern which informs the discussion, not any single number.

We agree with the reviewer that subject-level traces provide highly valuable information. This is the reason why we provide these traces in Supplemental Figures 2 and 3.

We were very cautious with introducing criteria for “mis-execution”. The aim of the investigation was to test the feasibility of the protocol, not to optimize the results. Hence the only type of “mis-execution” was a distinct deviation from the intended step size – a decision at the discretion of the investigator but supported in the straight-line condition by external marks.

Comment 2 (p. 3; also p. 9, p. 15). “Ready for use” reads a bit ahead of the data. You relied on clinicians imitating patients, and several planned conditions/sensors didn’t survive. Those two limitations should temper how broadly you pitch the conclusions.

Response 2: We adapted the conclusions such that the term “ready for use” is avoided. In particular, we use now the following phrasing:

Abstract, Conclusion:

“In conclusion, self-familiarization of health care professionals with detection limits of a physical activity tracker using specific protocols seems to be a feasible approach. Such protocols can provide valuable tools to facilitate the use of physical activity trackers in clinical applications. Additional research may allow further refinement of the protocol to generate input more comparable across participants and closer to the gait of patients.”

Outlook, first sentence:

“Our investigation illustrates the feasibility to self-familiarize health care professionals with detection limits of a physical activity tracker using specific protocols.”

Conclusion at the end of the paper:

“The type of protocol presented has the potential to inform HCPs about the ability of a PAT to detect low-impact steps that are characteristic of patients following knee surgery. The concept of self-familiarization may allow to develop valuable tools to facilitate the clinical application of PATs. Further research should focus on optimization of the protocol with respect to generate input more comparable across HCPs and closer to the gait of patients.”

Comment 3 (p. 4). Step size isn’t the same as “low-impact.” Impact is about cadence, velocity, and forces, not just geometry. Without direct cadence/speed or force thresholds, the label feels assumed rather than measured.

Response 3: We completely agree that step size is not the same as impact. We can see that the first sentence in the section “Elaboration of the protocol” can be misunderstood as equalling step size with impact and we have rephrased it accordingly:

“Walking speed is well known to influence a number of gait-related variables, and its  impact on a PAT’s ability to detect steps has been previously demonstrated.”

Note that  - a few sentences later – we explicitly say that low impact if characterized not only by step size.

We completely agree that the label “step size” refers to an intention, not a measurement. We have used the phrase “intended” already several times in the manuscript. For further clarification, we phrase the factor as “Step size intended” when describing the protocol at the start of Section 2.

Comment 4 (pp. 4–5). Running everyone through a fixed sequence invites learning, fatigue, and expectation effects. Randomization or counterbalancing would help separate true condition effects from order artifacts.

Response 4: We agree with the reviewer that randomization and counterbalancing would be useful techniques to estimate condition effects in an unbiased way. However, this is not the primary goal of this investigation. The conditions are simply used to generate variation in the exposure of the PATs.

In the Outlook section, we now point out that studies aiming at understanding conditions should follow more closely standards of experimental research:

“Corresponding studies should implement a stricter control of conditions and additional data collection (e.g. video recordings) than the present study, which focused on a proof of principle.”

Comment 5 (pp. 4–5). The turning and zig-zag tasks are underspecified. Path width, turning radius, and allowable angular speed matter; without them, between-person variability can swamp the contrasts you want to study.

Response 5: We agree with the reviewer that a detailed analysis of the effect of the conditions would benefit from a higher standardization. As said above, an investigation of the conditions was not a primary aim. The description “Turn by 90° after 10 steps; Zigzag-line: 90° shifts after 5, 10, and 15 steps; 360° Circle clockwise.” reflects a specification which can be explained verbally in a reasonable manner to participants.

Comment 6 (p. 4). The paper drifts from step counting into “activity profiles,” but the protocol never measures posture or non-ambulatory behavior. Tightening the scope to what you actually recorded would make the claims cleaner.

Response 6: We mentioned “activity profiles” only two times in the paper – once in the Introduction and once in the Discussion. We cannot see that we “drift from step counting into activity profiles”.

In the Introduction, we mention this as in clinical practice the interest will be often not in simple step counts, but in activity profiles – for example the time spend at a low, moderate or vigorous activity level. However, this depends in any case on correctly identifying steps. Hence it makes sense to focus on step counts in this paper.

In the Discussion, we exactly point out how the idea of self-familiarization can be extended to address physical activity patterns directly.

Comment 7 (pp. 4–6). Safety and ethics deserve a clearer line of sight. Rapid turns, zig-zags, and crutch use in a corridor have non-zero risk; a brief note on monitoring, near-misses, or adverse events would reassure readers.

Response 7: The ethical aspects are now addressed in the Discussion in a separate paragraph:

“As with any experimental protocol applied to humans, potential risks have to be balanced against the gain in information. Rapid turns, zig-zags, and the use of crutches is not without risk. However, as HCPs constitute the intended user group, it seems fair to expect that users can balance risk vs information gain individually.”

The absence of adverse events is now mentioned in the Results part:

“None of the participants reported any complaints or difficulties with the protocol and there were no adverse events.”

In addition, information on the frequency of the necessity to repeat a specific condition is given:

“In the first participant, the target step sizes could not be achieved in the first round and the instructor decided not to record the data. In addition, the instructor requested 19 out of 350 trials to be repeated.”

Comment 8 (p. 5). Multiple instructors and assistants add procedural noise you can’t model away. Either standardize more tightly or acknowledge and analyze rater effects; otherwise device differences may reflect who ran the trial.

Response 8: This was already mentioned as a limitation in the Discussion. (See also response 10).

Comment 9 (p. 5). The crutch condition is handled qualitatively—“partial weight bearing”—with no control of load, cadence, or placement. That makes the “walking-aids” factor hard to interpret: differences could be execution, not device.

Response 9: We agree with the reviewer. As said above, it was not a primary goal of the investigation to study the effect of the conditions.

Comment 10 (p. 6). Ground truth rests on one assistant’s manual count with no independent check. Even a light reliability pass (second rater or video spot-checks) would raise confidence in the reference.

Response 10: The limited control of the ground truth is now mentioned as a limitation:

“It cannot be excluded that the variation in instructors across the participants has contributed to the limited inter-HCP reproducibility. Similarly, the assessment of the ground truth by the assistant need not be error free, contributing to underestimation of reproducibility.”

Comment 11 (p. 6). Reading totals before/after and differencing is fragile. Transcription slips, display lag, or reset issues happen; describe checks to catch them or move to an automated log to harden the chain.

Response 11: We regard this as part of the limited control of the ground truth and refer to Response 10.

Comment 12 (pp. 6–7). The analysis hops between counts and ratios without a unifying plan. If ratios are the target, use models built for bounded outcomes (e.g., beta or hurdle) instead of forcing everything through a count model.

Response 12: We agree with the reviewer that the use of different models for slightly different versions of the outcome may add confusion. The reason for this is the unusual distribution of the ratio values. They can exceed 1.0, and they have a bimodal distribution with many values close to 0 or 1

For this reason, we tried to avoid working directly with the ratio values as numerical outcomes. Actually, only the computation of the ICC values is directly based on the ratio values – with the limitations mentioned in the manuscript. For the main analysis, we introduced a specific agreement rate taking the challenges implied by using the ratios into account.

With respect to comparing the conditions, we made use of a negative binomial model for count data. This is closely related to the idea of using ratios, as the number of steps performed added these models as fixed “exposure time”.  Hence the counts are always interpreted relative to the number of steps performed.

Consequently, we feel that the variation in analytical approach is well justified and less diverse as it may appear at first sight.

Comment 13 (p. 6). Protocol fidelity isn’t quantified. How often were targets missed, by how much, and how were those trials handled? Without that, it’s tough to separate condition effects from plain execution error.

Response 13: It is now explicitly mentioned in the result sections, how often the participants were asked to repeat a specific condition (cf. Response 7). This also clarifies that there are no single missing conditions in the analyses – except of the first round of the first participant which is missing completely.

Comment 14 (p. 6). Condition effects (e.g., crutches reducing trouser-sensor counts) don’t account for subject-level variability. A simple mixed model would help distinguish device behavior from individual strategy.

Response 14: Subject level variability was taken into account in the comparison of the conditions. However, this was not based on a mixed model but by using robust standard errors taking the clustering within each participant into account.

Stata offers a wide range of multi-level models, but to the best of our knowledge, zero inflated models are not covered. This limits the applicability of mixed models in this study.

Comment 15 (p. 6). The protocol assumes devices show immediate readable counts, which many trackers don’t. That practical constraint narrows applicability more than the text lets on—worth stating plainly.

Response 15: This was already mentioned in the Discussion.

Comment 16 (p. 7). The choice of a zero-inflated negative binomial for ~20-step trials is under-argued, and diagnostics are missing. Show why ZINB is preferred and that it fits, not just that it runs.

Response 16: Our choice of a ZINB is based on conceptual considerations – not on model comparisons. The idea of a detection limit suggests that a PAT produces either a 0 – because all steps are beyond the detection limit – or a count number reflecting how well the PAT can detect some of the 20 steps. Conditions may affect both components in a different manner. This is exactly the situation zero-inflated models have been developed for.

The choice of a negative binomial model (instead of a Poisson model) reflects exactly that we have to expect an influence of the participants. The expected count may vary from participant to participant. This would not be allowed in a Poisson model. 

We mention these arguments now in the Methods part:

“This model takes into account that conditions may influence the detection limit – implying potentially that no steps can be detected – and the ability to count steps above the limit correctly to a different degree and that participant effects introduce heterogeneity in counts.”

Goodness of fit tests for a ZINB would be of limited value due to the absence of more general models which are practically feasible.

Note that these models are only used to compute p-values for the comparison of different conditions. This does not reflect the primary aim of the study.

Comment 17 (p. 7). With many condition/category contrasts, some multiplicity control is expected. Otherwise a few “significant” hits may just be noise; even a transparent trade-off discussion would help.

Response 17: The p-values are only reported to give the reader an idea that the differences observed in Figure 3 do not reflect random noise and that the patterns observed make some sense – reflecting the expected major effect of step size and minor effects of the other conditions. No single p-value is interpreted as evidence for a specific, conceptual hypothesis. Hence, we do not regard multiplicity control as an essential issue. 

Comment 18 (p. 7). Agreement categories are borrowed and widely debated, and the metric lives mostly in the supplement. Define it clearly in the main text and justify thresholds with task-relevant reasoning.

Response 18: We followed the advice of the reviewer an describe the metric in more detail in the manuscript. We also comment on the choice of the thresholds 0.2 and 0.8:

“To address this issue, we consider also weighted agreement rates. In determining the degree of agreement between two single step count ratios, the following weights were used:

0.0 - If one step count ratio is above 0.8 and the other is below 0.2

0.5 - If one step count ratio is between 0.5 and 0.8 and the other is below 0.2 or if one step count ratio is between 0.2 and 0.5 and the other is above 0.8

1.0 If both step count ratios are between 0.2 and 0.8 or if one step count ratio is above 0.8 and the other above 0.5 or if one step count is below 0.2 and the other below 0.5.

The choice of the thresholds 0.2 and 0.8 reflects that even true ratios of 0 or 1 may be affected affected by some noise. The agreement rate of a set of pairs of step count ratios was then defined as the average weight over all pairs. As pointed out in Supplemental File 2, such agreement rates can be also transformed into   values following the principle introduced by Cohen[25].”

We kept Supplemental File 2, which still includes a more-in-depth justification.

Comment 19 (p. 7). Data sharing is good, but there’s no pre-specified analysis plan or code. Given custom metrics and nonstandard modeling, a small, documented workflow would make replication real.

Response 19: We provide now commented Stata code for the analyses presented in Sections 3.3. and 3.4.

Comment 20 (p. 9; also pp. 14–15). The conclusion leans toward endorsing a single sensor despite dropped conditions and failures elsewhere. If you keep that message, frame it as provisional and bound by the observed gaps.

Response 20: It is not our intention to endorse any single sensor. In contrast, we point out that even the sole sensor with “optimal” results requires further research and may lack practical feasibility.

Comment 21 (p. 9). Execution problems—knee-sensor failures, the missing slippers condition—aren’t minor. They change design balance and interpretability, and the discussion should grapple with that directly.

Response 21: We mention now the impact of these problems as a limitation:

“The failure to include the slipper condition implies a risk of overlooking a potential challenge to the PATs and the limited amount of data for the knee sensor may prevented further insights from a high quality, but still imperfect, sensor.”

Comment 22 (p. 10). Reproducibility concerns may be execution, not device. Without cadence or step-length checks, you can’t localize the instability, so claims about device inconsistency should be softened.

Response 22: The main point we try to make here is that the variation is unlikely to reflect an issue with the protocol.

With respect to execution, we feel that we have the strong point of paired observations. “Execution” was identical for the wrist and the trouser sensor. There may be some aspects specific for the wrist sensor, for example arm movements. However, it seems to us fair to regard this as “insufficient step counting quality of the sensor”, as from a clinical perspective it does not matter whether it is the sensor itself or its typical use, which causes the problem. To clarify this, we have added the phrase “in this specific clinical context”.

Comment 23 (pp. 10, 14). Clinical readiness is over-promised. Agreement is modest and inputs vary across people and conditions; “promising method development” is a better fit than “ready for use.”

Response 23: We agree with the reviewer, and we changed this accordingly. (Cf. Response 2)  

Comment 24 (pp. 10, 15). The sample-size story is thin: the target wasn’t met, precision goals aren’t stated, and heterogeneity is high. Emphasize interval estimates and acknowledge limited power.

Response 24: We added a section under limitations with respect to the sample size:

“The intended sample size of 16 HCPs was not reached. However, the sample size consideration aimed at ensuring that moderate differences between PATs with respect to detection limits could be demonstrated. Fortunately, the differences were rather distinct, and hence 14 HCPs were sufficient to obtain insights.”

As pointed out above, emphasizing interval estimation is challenging in this study.

Comment 25 (p. 14). Key device details are missing—firmware, sampling rates, filters, algorithm modes. Those parameters move step counts in practice; without them, comparisons and replications are guesswork.

Response 25: As we mainly use commercial sensors, we have to rely on the information provided by the manufacturers. The links in the Supplemental File 1 have been updated, and some information is added there. For the knee sensor, a reference to a recent publication is added.

Round 2

Reviewer 2 Report

Comments and Suggestions for Authors

The authors have effectively fulfilled the majority of the reviewer’s comments and requests, demonstrating substantial revisions to improve the manuscript.